

# Realistic scenarios of missing taxa in phylogenetic comparative methods and their effects on model selection and parameter estimation

Rafael S. Marcondes

Museum of Natural Science and Department of Biological Sciences, Louisiana State University, Baton Rouge, LA, United States of America

## ABSTRACT

Model-based analyses of continuous trait evolution enable rich evolutionary insight. These analyses require a phylogenetic tree and a vector of trait values for the tree's terminal taxa, but rarely do a tree and dataset include all taxa within a clade. Because the probability that a taxon is included in a dataset depends on ecological traits that have phylogenetic signal, missing taxa in real datasets should be expected to be phylogenetically clumped or correlated to the modelled trait. I examined whether those types of missing taxa represent a problem for model selection and parameter estimation. I simulated univariate traits under a suite of Brownian Motion and Ornstein-Uhlenbeck models, and assessed the performance of model selection and parameter estimation under absent, random, clumped or correlated missing taxa. I found that those analyses perform well under almost all scenarios, including situations with very sparsely sampled phylogenies. The only notable biases I detected were in parameter estimation under a very high percentage (90%) of correlated missing taxa. My results offer a degree of reassurance for studies of continuous trait evolution with missing taxa, but the problem of missing taxa in phylogenetic comparative methods still demands much further investigation. The framework I have described here might provide a starting point for future work.

## INTRODUCTION

Phylogenetic comparative biology is a thriving field that uses phylogenies towards the goal of elucidating the historical mechanisms that have given rise to species and their traits. At the core of phylogenetic comparative methods are statistical models that translate ideas about evolutionary processes into the language of mathematics, thus allowing biological explanations to be quantitatively weighed against one another, and their implications to be explored in detail (*Butler & King, 2004*; *Cressler, Butler & King, 2015*; *Brown & Thomson, 2018*; *Zuk & Travisano, 2018*). Model-based phylogenetic studies of univariate continuous trait evolution require at least two inputs: a phylogenetic tree of the species under study, and a vector of trait values for those species. One or more models are then typically fitted to the

Corresponding author
Rafael S. Marcondes,
raf.marcondes@gmail.com,
rmarco3@lsu.edu

trait vector, conditioned on the phylogeny. The fits of these models can then be compared (often using Akaike information criteria) to assess which one offers the best explanation of the data. In addition, parameter estimates from fitted models can give insights into features of the evolutionary process that generated the data, such as adaptive peaks, rates of evolution and strength of selection (*Beaulieu et al., 2012*; *O'Meara & Beaulieu, 2014*).

Phylogenies used in comparative studies are typically estimated from molecular sequences and trait data are typically generated from specimens deposited in natural history collections or, less often, from observations of live organisms. Because of differences in ecological characters such as range size, habitat preference, life history and behavior, not all taxa are equally likely to be available for inclusion in a molecular tree or trait dataset (*Garamszegi & Møller, 2011*). Therefore, comparative studies often have missing taxa, that is, taxa that are members of the clade under study, but which researchers have been unable to include in their analyses because they were either not included in the phylogeny or not accessible for measurement of traits (*Thomson & Shaffer, 2010*; *Slater et al., 2012*; *Slater, Harmon & Alfaro, 2012*; *Reddy, 2014*; *Rabosky, 2015*). This usually results in the missing taxa being excluded from the analyses, which has the potential to introduce biases in model selection and parameter estimation (*Garamszegi & Møller, 2011*; *Pennell, FitzJohn & Cornwell, 2016*).

Missing data have received significant attention in the context of nucleotide sequences used in phylogeny estimation (e.g., *Wiens & Morrill, 2011*; *Jiang et al., 2014*; *Eaton et al., 2017*). In stark contrast, missing data have received little attention in phylogenetic comparative biology (but see *Garamszegi & Møller, 2011*), and, in the rare occasions when researchers assessed impacts of missing taxa in comparative models, they were simulated in a random fashion with respect both to trait values and to phylogeny (e.g., *Ingram & Mahler, 2013*). In the next few paragraphs, I argue that missing taxa are unlikely to occur randomly and describe more realistic scenarios where missing taxa are phylogenetically clumped and/or correlated to the trait of interest. Next, I describe a set of simulations where my aims were two-fold: first, to conduct an initial, exploratory investigation of the impacts of realistic missing taxa on a limited set of models of univariate continuous trait evolution; and, second, by presenting a basic framework that can be taken up by other investigators, to instigate more attention and future research on this neglected issue in phylogenetic comparative biology.

## Realistic scenarios of missing taxa in comparative datasets

The multifarious ecological traits that influence the probability that a taxon is sampled have phylogenetic signal. Consequently, missing taxa should be expected to be phylogenetically clumped. The most important ecological trait influencing sampling probability is rarity, broadly understood as the character of a taxon that has a low abundance and/or a small geographical distribution (*Gaston, 1994*). It is intuitive that rarer taxa are more difficult to detect, observe, capture and collect, and thus more likely to be missing from datasets than common taxa. Both components of rarity, abundance and range size, have been shown to have phylogenetic signal, i.e., they are heritable at a macroevolutionary level. For example, 41% of the variation in population density of North American birds can be attributed to

their taxonomic family (*Maurer, 1991*), and taxonomic affiliation also explains variation in abundance among Neotropical rainforest mammals (*Arita et al., 1990*). As for range size, it has been shown to be heritable in studies of mammals (*DeSantis et al., 2012*), birds (*Waldron, 2007*; *Herrera-Alsina & Villegas-Patraca, 2014*), mollusks (*Jablonski, 1987*) and herbaceous plants (*Qian & Ricklefs, 2004*). That rarity has phylogenetic signal was also indicated by *Fritz & Purvis (2010)* finding that threatened species of British birds and of the world's mammals tended to be phylogenetically clumped.

Beyond rarity, various other axes of an organism's ecological niche can affect its probability of being sampled in comparative studies, and the pervasiveness of phylogenetic niche conservatism (*Losos, 2008*; *Wiens et al., 2010*; *Crisp & Cook, 2012*) is evidence of the high phylogenetic signal of ecological niches. Because taxa inhabiting climates that are hostile to humans (for example, boreal organisms; *Malaney & Cook, 2018*) are less likely to be sampled, climate is a particularly important niche axis in this context, and it has been repeatedly shown to display phylogenetic signal (reviewed by *Wiens & Graham, 2005*). Another important niche axis, habitat preference, also has phylogenetic signal (*Barr & Scott, 2014*) and is likely to affect sampling probability, because forest-based organisms are more difficult to detect and collect than nonforest organisms.

These taxonomic biases in rarity presumably translate into taxonomic biases in availability of specimens and genetic samples. For example, *Malaney & Cook (2018)* reported strong taxonomic imbalance in collections of North American mammals, with Rodentia being comparatively overrepresented in relation to all other mammal orders, and *Reddy (2014)* described similar imbalances for the availability of genetic data for the world's birds.

In addition to being phylogenetically clumped, missing taxa might sometimes be directly correlated to the trait under study, when taxa with a higher (or lower) trait value have a lower probability of being sampled. For example, because taxa with small range sizes are more difficult to sample, they will be missing from molecular datasets more often than taxa with large ranges (*Reddy, 2014*). This means that unsampled taxa will have a smaller average range size than sampled taxa. Therefore, if we were undertaking a comparative study of range size with incomplete taxon sampling, the distribution of trait values in our sample will differ from the real distribution, potentially biasing model-based analyses. This should also be the case for other ecological traits tightly linked to rarity and sampling probabilities, such as abundance and some niche axes.

In sum, there is ample evidence of phylogenetic signal in ecological traits likely to influence sampling probability for comparative studies. Consequently, sampling probability itself should display phylogenetic signal, and missing taxa are likely to be phylogenetically clumped. There is also reason to expect some types of traits to be directly correlated to sampling probabilities. Here, I present simulation-based analyses examining how those realistic scenarios of missing taxa might affect the performance of models of continuous trait evolution.

**Table 1  Summary of simulated traits and their generating models.**

| Abbreviation of trait | Description | Model |
|---|---|---|
| R | Binary trait representing regimes required for BMS and OUM models for **T** | Equal-rates MK model |
| T | Continuous trait about whose evolution we are interested in making inferences | BM, BMS, OU or OUM |
| S | Binary trait determining the sampling status of each tip (0 = missing, 1 = sampled) | Threshold model for cluMT; non-phylogenetic for corMT |
| L | Continuous trait representing the liability underlying **S** in the threshold model in the cluMT scenario | BM |

# METHODS

## Overview

My simulations were designed to examine how realistic scenarios of missing taxa affect model selection and parameter estimates for a single continuous trait in whose evolution we are interested, hereafter referred to as **T** (Table 1). I simulated **T** under a number of different models and then pruned 10%, 50% or 90% of terminal taxa from the tree under three different schemes (Fig. 1): (1) randomly (rMT); (2) phylogenetically clumped missing taxa (cluMT); and (3) correlated to **T** (corMT). For comparison, I also ran simulations with no missing taxa (nMT). The sampling status of a taxon can be thought of as a binary character, hereafter referred to as **S** (Table 1), with states 0 (missing) and 1 (sampled). Once **T** and **S** were simulated, I pruned tips from the tree and dataset based on **S** and, finally, I fitted all models to **T** and assessed support for the generating model, as well as precision and bias of parameter estimates. I repeated the simulation 1000 times for each combination of type of missing taxa, percentage of missing taxa, and model of **T** trait evolution.

Before presenting the details of my simulations, I reiterate that I did not seek to exhaust every possible scenario of missing taxa in phylogenetic comparative biology. Rather, I sought to explore the implications of two very specific scenarios that I argue are likely in real datasets. There undoubtedly exist other possible, if less probable, configurations of missing taxa, and they may have different impacts on model performance, but addressing those was beyond the scope of my study. I also acknowledge that, even within the scenarios I studied, some of my simulation settings necessarily entailed some degree of arbitrariness, for example in the size of the trees or in the parameter values I used to simulate various traits. Different choices might have led to different results, but while exploring those choices would certainly be productive, it was not my present objective in this initial, exploratory study.

## Details

I started each simulation by generating a random phylogenetic tree under a pure-birth model using the function *rphylo* in the R package *ape* (*Paradis & Schliep, 2019*) and rescaling the tree to unit height using the function *rescale* in the R package *geiger* (*Pennell et al., 2014*). The size of each initial tree was set so that the number of tips *after* dropping missing taxa was always equal to 300. For example, when simulating 10% missing taxa,

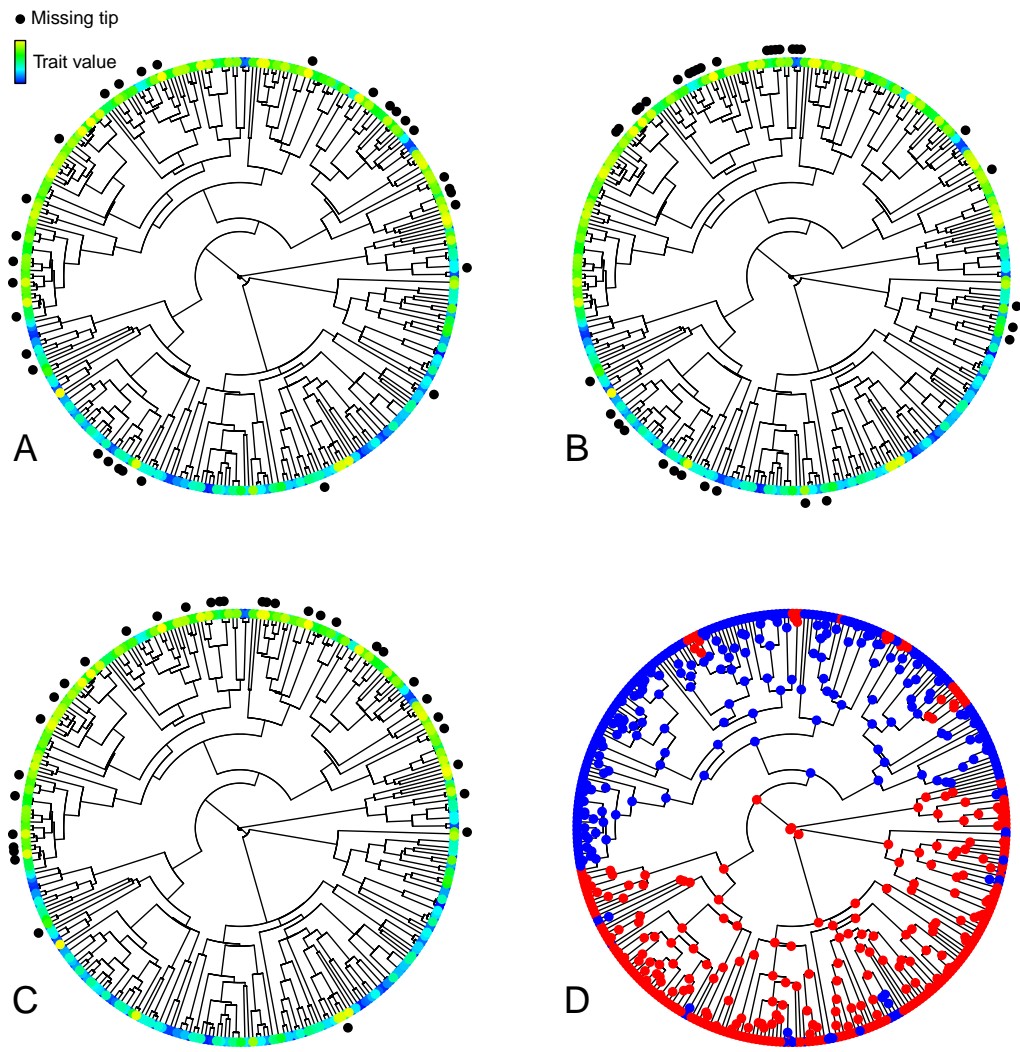

**Figure 1** **Illustration of the various simulated configurations of missing taxa.** (A–C) Trees depicting a trait **T** simulated under an OUM model, and random (A), phylogenetically clumped (B), or correlated (C) missing taxa. (D) Selective regimes, simulated under an MK model, underlying variation in the theta parameter in the OUM model used to simulate **T** in A–C. The tree in this figure contains 333 tips, 33 of which were simulated to be missing in each panel, thus corresponding to a 10% missing taxa scenario.

the initial number of tips was 333, and when simulating 50% missing taxa it was 600. This ensured that my results were affected only by missing taxa *per se*, and not by tree size, which is known to affect the performance of models of trait evolution (*Beaulieu et al., 2012*; *Boettiger, Coop & Ralph, 2012*).

To represent regimes underlying variation in BMS and OUM parameters (see below), I simulated the evolution of a binary trait **R** under an equal rates MK model with a transition rate $q = 0.5$ (Table 1, Fig. 1D), and I ensured that the smallest of the two regimes always included no fewer than 25% and no more than 45% of tips *after* dropping the tips
representing missing taxa. To simulate this trait, I used the function *simulate_mk_model* in the R package *castor* (*Louca & Doebeli, 2017*).

After simulating the tree and **R**, I simulated **T** (Table 1), the continuous trait of interest, under one of four models using the function *OUwie.sim* in the R package *OUwie* (*Beaulieu et al., 2012*): single-rate Brownian Motion (BM), single-optimum Ornstein–Uhlenbeck (OU), multiple-rate Brownian Motion (BMS), and multiple-optimum Ornstein–Uhlenbeck (OUM). Under BM, changes in trait value are purely nondirectional and governed by a single parameter, $\sigma^2$ (sigma-square), that determines the rate of evolution (*Felsenstein, 1985*). Under OU, trait evolution is controlled by a nondirectional component represented by $\sigma^2$ as well as by a directional component under which trait values change preferentially towards an optimum ($\theta$, theta) with strength of attraction $\alpha$ (alpha) (*Hansen, 1997*). BMS and OUM represent variations of BM and OU in which $\sigma^2$ and $\theta$, respectively, are allowed to assume different values depending on regimes (trait **R** in my simulations) reconstructed *a priori* on the phylogeny (*Butler & King, 2004*; *O'Meara et al., 2006*; *Beaulieu et al., 2012*).

For each model, I set the rate parameter $\sigma^2$ at the root ($\sigma_0^2$) to a value of 0.5, and only for the BMS model it shifted to $\sigma_1^2 = 1$ in the derived regime. For the OU and OUM models, I set the optimum parameter $\theta$ to 10 at the root ($\theta_0$), with a shift to $\theta_1 = 11$ in the derived regime under OUM. The $\alpha$ parameter of OU and OUM models, representing the strength of attraction to the optimum, was always constant at 1.5. This $\alpha$ value corresponds to a phylogenetic half-life (the time, as a proportion of the tree height, that a trait takes to evolve halfway towards the adaptive peak $\theta$) of 0.46, thus representing an OU process of moderate strength.

For the cluMT scenario (Fig. 1B), I simulated **S**, the trait determining the sampling status of each tip (Table 1), under a threshold model (*Felsenstein, 2005*; *Fritz & Purvis, 2010*), where **S** is underlain by a continuous liability trait **L** (Table 1). The state of **S** for each tip depends exclusively on whether **L** is above or below a certain threshold value. Because sampling status is likely to be a highly complex trait determined by innumerable neutral and adaptive evolutionary forces, its evolution should resemble a purely nondirectional process over macroevolutionary time and be best described by a simple Brownian Motion model (*O'Meara et al., 2006*), which I thus chose to simulate **L**. I used a $\sigma^2$ value of 1 in the function *OUwie.sim*. The threshold for **S** was set based on the desired percentage of missing taxa, so that, for instance, when simulating 10% missing taxa, the tips with the 10% lowest values of **L** were assigned state 0 and the tips with the 90% highest values were assigned state 1.

For corMT (Fig. 1C), I simulated **S** non-phylogenetically, using the R native function *sample.int* to sample tips to be dropped. I provided that function with an integer corresponding to the total number of tree tips (argument *n*), the desired number of tips to be dropped (*size*), and a vector of weights for obtaining the elements of the vector being sampled (*prob*). Those weights were calculated, for each tip, as:

$$w = \frac{t}{\text{sum}(T)} - \frac{\min(T)}{\text{sum}(T)},$$

where *t* is the value of the trait of interest for that tip, and T is the vector of *t* values for all tips. *Sample.int* then used those weights to sample *size* integers from the interval 1:*n*. The sampled integers correspond to tips to drop. This procedure results in the sampling probability of each tip being linearly proportional to its **T** value. The tip with the lowest **T** value had a 100% chance of being sampled, and tips with a higher **T** value were probabilistically more likely, but not certain, to be missing (Fig. 1C).

Due to the different ways in which missing tips were sampled, in the cluMT and corMT scenarios the number of missing tips always corresponded exactly to the desired proportion, but in the rMT case, that number varied slightly due to the probabilistic sampling. For example, under 50% rMT the actual proportion of missing tips varied from 0.43 to 0.57. However, this does not bias my inferences because over 1,000 simulations the number of missing tips will average to the desired proportion, and the results will reflect that.

For each simulation in each scenario of missing taxa, I quantified the phylogenetic signal in **S** using *Fritz & Purvis (2010)* D statistic, calculated with the function *phylo.d* in *caper* (*Orme et al., 2017*). D equals 0 when a binary trait has evolved under a threshold model with a Brownian liability as described above, and 1 when it has a phylogenetically random distribution at the tips of the tree.

Once **T** had been simulated and tips had been pruned based on **S**, I used *OUwie* to fit each of the four models to **T** and assess their support using sample size-corrected Akaike information criteria (AICc; *Burham & Anderson, 1998*), under the expectation that the generating model should have the lowest AICc score. For each combination of generating model, scenario of missing taxa, and percentage of missing taxa, I also calculated the median delta AICc of the generating model in the set of simulations in which it was not the top model. Delta AICc equals the AICc score of the focus model minus the AICc score of the model with the most support (lowest AICc), and is thus a measure of relative support of a model compared to the top model.

Finally, I computed the bias and precision of parameter estimates under each scenario to assess how they were affected by missing taxa. I calculated bias as the mean parameter estimate minus the generating parameter value. I calculated precision as the median absolute deviation (median deviation from the median) of parameter estimates among simulations. I used this statistic in lieu of the simple variance because it is more robust to outliers. I normalized both bias and precision by dividing them by the generating parameter values.

## RESULTS

The mean value of Fritz and Purvis' D statistic for **S**, the trait determining the sampling status of each tip, was 1.004 and −0.019 across all simulations in the rMT and cluMT scenarios respectively, indicating, as intended, low phylogenetic signal of missing taxa in the former, and high phylogenetic signal of missing taxa in the latter. For corMT, because **S** was correlated to **T**, its phylogenetic signal was computed separately depending on **T**'s model of evolution. The mean value of Fritz and Purvis' D statistic in that scenario was 0.921, 0.931, 0.944, and 0.932 for **T** simulated under BM, BMS, OU and OUM, respectively.

**Table 2 Model selection error rates.** Number of simulations, for each combination of type of missing taxa, percentage of missing taxa, and generating model for the trait of interest T, in which the model with the greatest support (lowest AICc) was not the generating model. The numbers are always out of 1,000 replicated simulations.

| | | BM | BMS | OU | OUM |
|---|---|---|---|---|---|
| No missing taxa | | 306 | 61 | 181 | 22 |
| | 10% | 296 | 53 | 176 | 13 |
| Random missing taxa | 50% | 298 | 70 | 161 | 11 |
| | 90% | 266 | 83 | 196 | 4 |
| | 10% | 279 | 63 | 167 | 18 |
| Clumped missing taxa | 50% | 291 | 63 | 195 | 22 |
| | 90% | 318 | 60 | 194 | 24 |
| | 10% | 300 | 47 | 178 | 25 |
| Correlated missing taxa | 50% | 305 | 73 | 177 | 16 |
| | 90% | 381 | 99 | 169 | 11 |

Clumped and correlated missing taxa resulted at most in a slight increase in the model selection error rate compared to no missing taxa or to random missing taxa, as indicated by the number of simulations in which the generating model did not have the lowest AICc among the four models (Table 2, Fig. 2). For example, when the generating model was OUM, it failed to receive the most support in 22 out of 1,000 simulations under a no missing taxa scenario, 4 out of 1,000 under 90% rMT, 24 out of 1,000 under 90% cluMT, and 11 out of 1,000 under 90% corMT. The model selection error rate did not consistently increase with the percentage of missing taxa under any scenario, but under all scenarios the error rate was higher (always ∼30%) when BM was the generating model, usually followed by OU, then by BMS and OUM (Table 2, Fig. 2). BM was most often confused for BMS and OU, and more rarely confused for OUM (Fig. 2). Likewise, BMS was confused more often for BM and OU than for OUM (Fig. 2). In contrast, OU was often confused for OUM. Finally, OUM was only very rarely not selected as the top model when it was the generating model (Table 2, Fig. 2), indicating that it leaves the strongest signature on the data among all models examined here.

Further revealing that missing taxa have little impact on model selection, the median delta AICc of the generating model in the set of simulations where it was not the top model was almost always ≤2 (Table 3). This means, following Burham and Anderson's (1998) rule of thumb that "models having delta AICc ≤ 2 have substantial support", that the generating model still had relatively high support even in the cases where it was not the top model.

I also examined the bias (Table 4) and precision (Table 5) of parameter estimates across simulations in each scenario. Because I normalized these metrics, they are directly comparable across parameters. The absolute bias (Table 4) of $\sigma_0^2$ was always low (<0.0374) for all models and almost always remained so for $\sigma_1^2$ in BMS, even though $\sigma_1^2$ covered much smaller proportions of my simulated trees. The absolute bias of $\theta_0$ also remained fairly low (< 0.01), but the absolute bias of $\theta_1$ was noticeably higher across most situations (often > 0.2). Finally, $\alpha$ had by far the greatest bias of all parameters (almost always > 0.1).

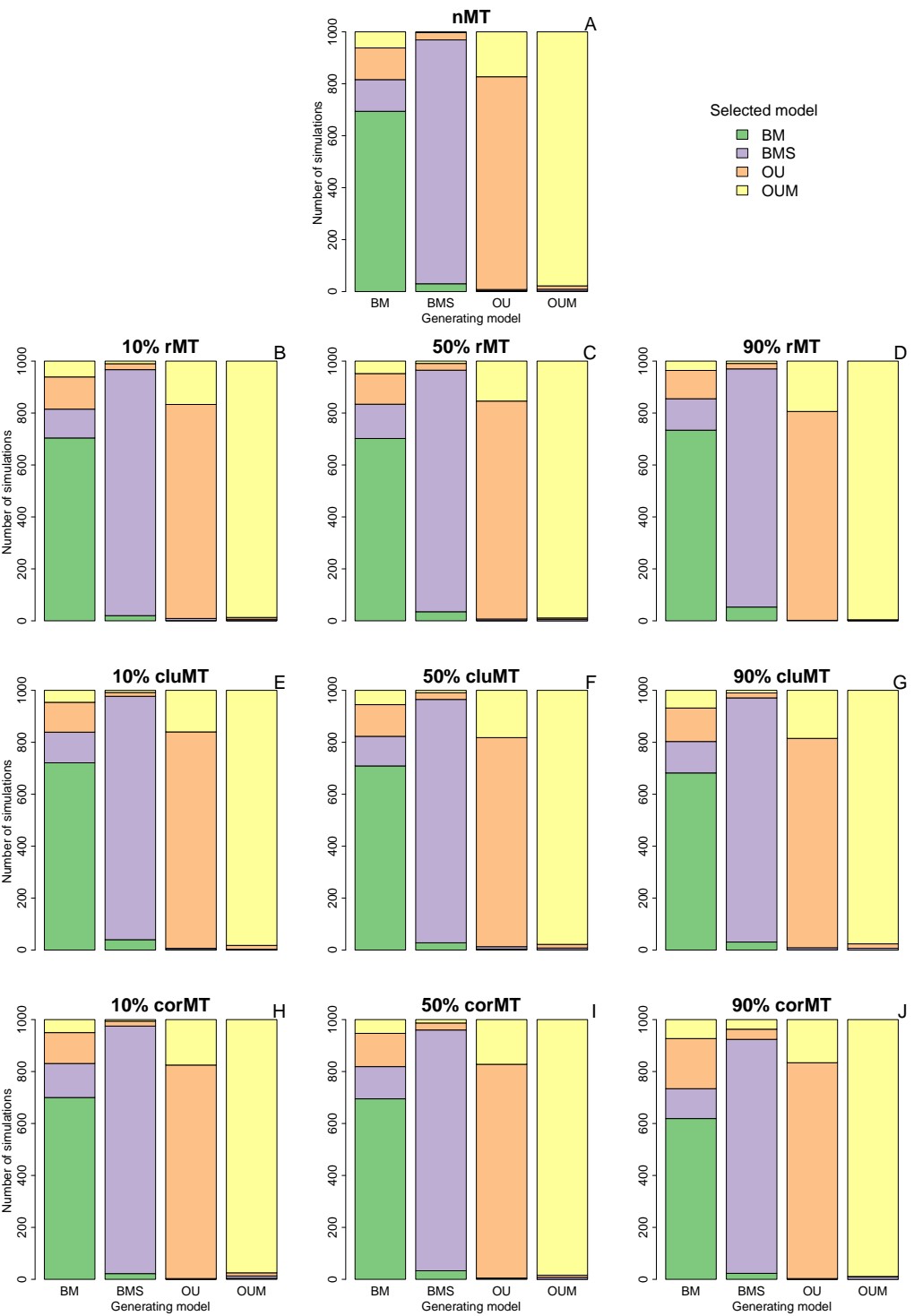

**Figure 2  Best-fitting models selected in each set of simulations.** Bar plots depicting the number of simulations in which each model was selected by AICc as the best fitting model, under each combination of type of missing taxa, percentage of missing taxa, and trait of interest generating model. Types of missing taxa: nMT, no missing taxa; rMT, random missing taxa; cluMT, phylogenetically clumped missing taxa; corrMT missing taxa correlated to the trait of interest.

**Table 3  Delta AICc of the generating model when it was not selected as the top model.** Median delta AICc of the generating model for the trait of interest **T** in the set of simulations, under each combination of type of missing taxa, percentage of missing taxa and generating model, in which the trait of interest was not the model with the greatest support (lowest AICc).

|  |  | BM | BMS | OU | OUM |
|---|---|---|---|---|---|
| No missing taxa |  | 1.592 | 2.127 | 1.422 | 1.791 |
| Random missing taxa | 10% | 1.676 | 1.471 | 1.215 | 1.826 |
|  | 50% | 1.759 | 1.927 | 1.459 | 25.669 |
|  | 90% | 1.577 | 1.508 | 1.391 | 4.261 |
| Clumped missing taxa | 10% | 1.966 | 1.784 | 1.047 | 1.370 |
|  | 50% | 1.866 | 2.174 | 1.196 | 2.325 |
|  | 90% | 1.811 | 2.079 | 1.210 | 1.515 |
| Correlated missing taxa | 10% | 1.865 | 2.364 | 1.414 | 1.364 |
|  | 50% | 1.621 | 1.442 | 1.243 | 3.016 |
|  | 90% | 2.252 | 2.353 | 1.441 | 3.875 |

**Table 4  Bias of parameter estimates from models fitted to data simulated under various scenarios and proportions of missing taxa.** Bias was calculated as the generating parameter value, minus the mean estimated parameter across 1,000 simulations, divided by the generating parameter value.

| Parameter | Model | Normalized bias per scenario | | | | | | | | | |
|---|---|---|---|---|---|---|---|---|---|---|---|
|  |  | nMT | 10% rMT | 50% rMT | 90% rMT | 10% cluMT | 50% cluMT | 90% cluMT | 10% corMT | 50% corMT | 90% corMT |
| $\sigma_0^2$ | BM | −0.0084 | −0.0060 | −0.0006 | −0.0062 | −0.0002 | −0.0014 | −0.0012 | −0.0032 | −0.0090 | −0.0374 |
| $\sigma_0^2$ | BMS | −0.0100 | −0.0040 | −0.0080 | −0.0060 | −0.0060 | −0.0120 | −0.0080 | −0.0080 | −0.0120 | −0.0300 |
| $\sigma_1^2$ | BMS | −0.0090 | −0.0180 | −0.0080 | −0.0150 | −0.0090 | −0.0080 | 0.0000 | −0.0070 | −0.0240 | −0.0550 |
| $\sigma_0^2$ | OU | 0.0280 | 0.0280 | 0.0240 | 0.0400 | 0.0120 | 0.0240 | 0.0240 | 0.0240 | 0.0220 | 0.0280 |
| $\theta_0$ | OU | 0.0002 | −0.0004 | 0.0001 | 0.0002 | −0.0003 | 0.0001 | 0.0004 | 0.0000 | −0.0056 | −0.0217 |
| $\alpha$ | OU | 0.1073 | 0.0907 | 0.0807 | 0.0853 | 0.0680 | 0.0873 | 0.0887 | 0.1107 | 0.0887 | 0.1460 |
| $\sigma_0^2$ | OUM | 0.0460 | 0.0440 | 0.0560 | 0.1040 | 0.0520 | 0.0420 | 0.0340 | 0.0500 | 0.0560 | 0.0720 |
| $\theta_0$ | OUM | 0.0033 | 0.0034 | 0.0029 | 0.0032 | 0.0034 | 0.0036 | 0.0036 | 0.0025 | −0.0011 | −0.0142 |
| $\theta_1$ | OUM | −0.0220 | −0.0215 | −0.0183 | −0.0132 | −0.0215 | −0.0222 | −0.0200 | −0.0231 | −0.0261 | −0.0413 |
| $\alpha$ | OUM | 0.1347 | 0.1207 | 0.1207 | 0.1347 | 0.1400 | 0.1280 | 0.1040 | 0.1473 | 0.1280 | 0.1600 |

**Notes.**
Types of missing taxa: nMT, no missing taxa; rMT, random missing taxa; cluMT, phylogenetically clumped missing taxa; corrMT missing taxa correlated to the trait of interest.

Notably, although biases were generally low, they often tended to be systematic, meaning they were always in the same direction. $\sigma^2$ was consistently underestimated in BM and BMS models and overestimated in OU and OUM. $\alpha$ was always overestimated, and $\theta_1$ always underestimated. Only for $\theta_0$ was bias not systematic, as it was sometimes positive and sometimes negative. Regarding precision of parameter estimates, across all scenarios it tended to be to be better (meaning parameter estimates were more concentrated around the generating parameter value) for $\theta$, followed by $\sigma^2$, and worst for $\alpha$ (Table 5).

Interestingly, looking at each parameter individually, both bias and precision were largely similar across almost all scenarios, revealing that missing taxa had little effect on parameter estimation. Only at 90% corMT did the bias of most parameters worsen relative

**Table 5  Precision of parameter estimates from models fitted to data simulated under various scenarios and proportions of missing taxa.** Precision was calculated as the median deviation from the median estimated parameter across 1,000 simulations, divided by the generating parameter value.

| Parameter | Model | Normalized precision per scenario | | | | | | | | | |
|---|---|---|---|---|---|---|---|---|---|---|---|
| | | nMT | 10% rMT | 50% rMT | 90% rMT | 10% cluMT | 50% cluMT | 90% cluMT | 10% corMT | 50% corMT | 90% corMT |
| $\sigma_0^2$ | BM | 0.0800 | 0.0768 | 0.0838 | 0.0782 | 0.0792 | 0.0816 | 0.0810 | 0.0862 | 0.0756 | 0.0834 |
| $\sigma_0^2$ | BMS | 0.1080 | 0.1000 | 0.1040 | 0.1040 | 0.1080 | 0.1000 | 0.1120 | 0.1020 | 0.1020 | 0.1020 |
| $\sigma_1^2$ | BMS | 0.1340 | 0.1380 | 0.1420 | 0.1440 | 0.1430 | 0.1420 | 0.1400 | 0.1350 | 0.1460 | 0.1460 |
| $\sigma_0^2$ | OU | 0.1180 | 0.1180 | 0.1160 | 0.1500 | 0.1240 | 0.1140 | 0.1140 | 0.1220 | 0.1220 | 0.1460 |
| $\theta_0$ | OU | 0.0090 | 0.0084 | 0.0079 | 0.0068 | 0.0085 | 0.0090 | 0.0088 | 0.0082 | 0.0082 | 0.0071 |
| $\alpha$ | OU | 0.2893 | 0.2587 | 0.2727 | 0.2520 | 0.2840 | 0.2673 | 0.2580 | 0.2673 | 0.2567 | 0.2547 |
| $\sigma_0^2$ | OUM | 0.1260 | 0.1220 | 0.1360 | 0.1660 | 0.1160 | 0.1200 | 0.1180 | 0.1300 | 0.1260 | 0.1580 |
| $\theta_0$ | OUM | 0.0106 | 0.0090 | 0.0089 | 0.0077 | 0.0102 | 0.0098 | 0.0094 | 0.0090 | 0.0089 | 0.0078 |
| $\theta_1$ | OUM | 0.0141 | 0.0145 | 0.0147 | 0.0144 | 0.0141 | 0.0149 | 0.0160 | 0.0135 | 0.0134 | 0.0108 |
| $\alpha$ | OUM | 0.2893 | 0.2760 | 0.2727 | 0.2847 | 0.2813 | 0.2787 | 0.3027 | 0.3147 | 0.2953 | 0.2873 |

**Notes.**

Types of missing taxa: nMT, no missing taxa; rMT, random missing taxa; cluMT, phylogenetically clumped missing taxa; corrMT missing taxa correlated to the trait of interest.

to other scenarios and percentages (Table 4, Fig. 3). For example, the bias of $\sigma_0^2$ in the BM model was $-0.0084$ under nMT, $-0.0067$ under 90% rMT, $-0.0012$ under 90% cluMT, and jumped to $-0.0374$ under 90% corMT. Similarly, the bias of $\theta_0$ in the OUM model was $-0.0033$ at nMT, $-0.0032$ at 90% rMT, $-0.0036$ at 90% cluMT, and much greater at $-0.0142$ at 90% corMT. In contrast, the precision of parameter estimates did not deteriorate consistently even under 90% corMT (Table 5). For example, for $\sigma_0^2$ in the BM model, the precision was 0.0834 under 90% corMT compared to 0.0800 under nMT and 0.0862 under 10% corMT.

# DISCUSSION

## Model performance in the presence of missing taxa

Random and phylogenetically clumped missing taxa had little effect on model selection and parameter estimation for univariate continuous traits evolving along phylogenetic trees, even when 90% of taxa in a tree were missing. This result can be intuitively understood by appreciating that neither of these two scenarios change the distribution of trait values at the tips of the tree. Under random missing taxa, each tree tip has the same chance of being missing, independent of its trait value or phylogenetic position. Under phylogenetically clumped missing taxa, the chances of a tip being missing are independent of trait values, meaning that if a taxon is missing, it is likely that its close relatives will also be missing. These close relatives probably have similar trait values, but the distribution of trait values at the tips will not change because clumps of missing taxa will be evenly distributed across the tree, and trait values are not correlated across these clumps (Fig. 1). The only notable effect of missing taxa was when 90% of taxa were missing under a scenario in which sampling probability was correlated to the trait of interest (corMT). In that situation, there was a noticeable deterioration in the bias, but not the precision, of all parameters. This result

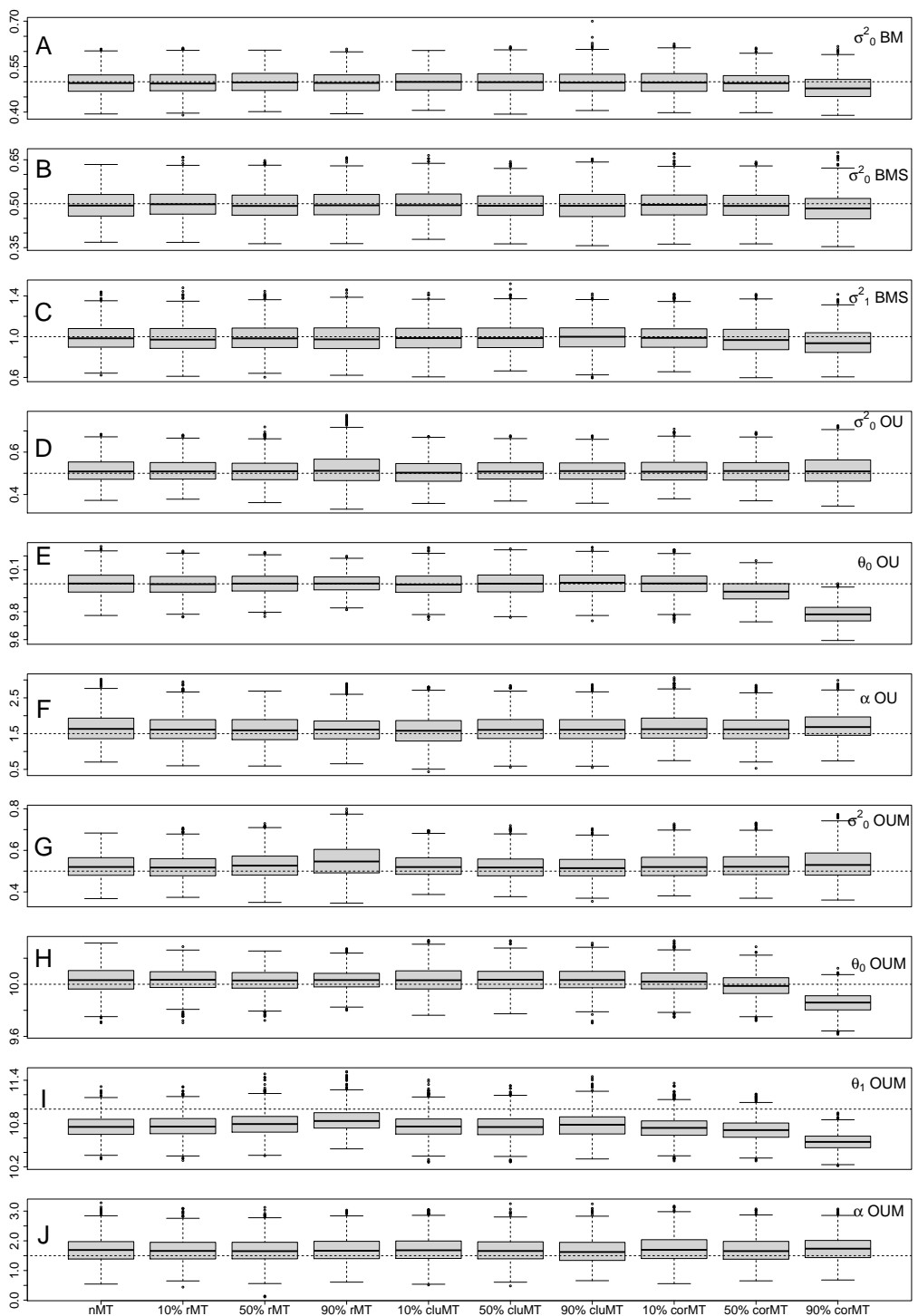

**Figure 3** **Distribution of estimated parameter values for each parameter in each combination of type of missing taxa, percentage of missing taxa and trait of interest generating model.** The dotted lines represent the real, generating parameter values. The type and percentage of missing taxa are indicated at the bottom of the figure, whereas the model and parameters are shown at the right-hand edge of the figure.Types of missing taxa: nMT, no missing taxa; rMT, random missing taxa; cluMT, phylogenetically clumped missing taxa; corrMT missing taxa correlated to the trait of interest.

can be understood by realizing that this is the only type of missing taxa that alters the distribution of trait values at the tips, thus misleading estimates.

## Model performance without missing taxa

Even though my main interest was exploring the implications of missing taxa, and those proved to be minor, it is also valuable to discuss performance in a scenario without missing taxa, seeing as model performance, in particular of OU models, has received significant recent attention in the form of simulation studies (*Beaulieu et al., 2012*; *Ho & Ané, 2014*; *Cooper et al., 2016*; *Cressler, Butler & King, 2015*).

BM was by far the model that most often failed to be selected by AICc when it was in fact the generating model, with an error rate of ∼30% across all scenarios (Table 2, Fig. 2). This suggests that BM is quite prone to mimicking patterns expected under other models, echoing *Slater, Harmon & Alfaro*'s (*2012*) finding, in a simulated birth-death phylogeny with 100 extant tips, that statistical power is low to favor BM over other candidate models even when it is the generating model. In particular, in my simulations with no missing taxa, a single-peak OU model was mistakenly selected over BM in 12.2% of the simulations (Fig. 2) when BM was the generating model. This result can be compared to *Cooper et al.*'s (*2016*), who similarly simulated BM datasets and computed how often an OU model was mistakenly selected as the best-fitting model. However, instead of using Akaike information criteria for model selection, they used likelihood-ratio tests and Bayes factors. The error rate they found was considerably lower than what I found using AICc, for example 5.5% and 0.2% in pure-birth trees with 200 tips using likelihood-ratio tests and Bayes factors respectively.

To explore the discrepancy between my and *Cooper et al.*'s (*2016*) results, I ran likelihood ratio tests (LRTs) on the results of my nMT simulations under BM (details presented as Supplemental Information). I found that LRTs incorrectly preferred OU over the true BM model in 6.7% of the simulations, a number much lower than the AICc error rate of 12.2% and more in line with *Cooper et al.*'s (*2016*) 5.5%. This result suggests that LRTs may perform better than AICc as a tool to discriminate between BM and OU models in comparative datasets, an issue that, while beyond the purview of the present paper, deserves further investigation.

Regarding parameter estimates, $\alpha$ was the hardest parameter to estimate in every scenario, seeing as it had the largest biases and poorest precisions, sometimes by orders of magnitude, compared to all other parameters (Tables 4 and 5). This result is consistent with *Cressler, Butler & King*'s (*2015*), who found the related parameter $\eta$ (selection opportunity, equaling the product of $\alpha$ and the height of the phylogeny) the hardest parameter to estimate, and $\theta$ the easiest in OUM models. In my results $\alpha$ and $\sigma^2$ estimates tended to have a consistent upward bias in OUM models, whereas $\theta_1$ was biased downwards and $\theta_0$ did not have a consistent upwards or downwards bias (Fig. 3). These results are similar to *Beaulieu et al.*'s (*2012*), except, interestingly, for $\theta_0$, which *Beaulieu et al. (2012)* found to be underestimated as well, although in that case simulations were based on more complex models.
## Broader implications for models of trait evolution in the presence of missing taxa

My results bear on the issue of birth-death polytomy resolver (BDPR) algorithms, such as those of *Kuhn, Mooers & Thomas (2011)* and *Thomas et al. (2013)*, that impute the position of missing taxa onto a phylogeny based solely on taxonomic information and a diversification model. *Jetz et al. (2012)* used a BDPR to add unsampled taxa to a species-level phylogeny of the world's birds, amounting to 33% of all species in the final tree. This approach was shown by *Rabosky (2015)* to systematically and severely bias downstream analyses of trait macroevolution. For example, when using a BDPR algorithm, *Rabosky (2015)* found the Brownian rate parameter $\sigma^2$ to be overestimated by an average factor of 3.51 times, compared to a bias of no more than 0.02 in any of my scenarios with missing taxa (Table 4). Therefore, my results suggest that the reliability of BDPR algorithms may be a moot point for studies concerned mainly with continuous traits, because model selection and parameter estimation might perform better on phylogenies with a very high proportion of realistic missing taxa than on BDPR-imputed phylogenies. This is in accordance with *Rabosky*'s (*2015*) "genes only" scenario, where he ran trait evolution analyses with minimal bias after pruning BDPR-imputed species from the *Jetz et al. (2012)* tree. However, this is not to say that BDPR algorithms are never useful. We need to remember that BDPRs do demonstrably improve the performance of comparative studies that do not include traits, such as speciation-extinction analyses (*Rabosky, 2015*).

Even though my and *Rabosky*'s (*2015*) findings offer some reassurance for the use of incompletely sampled phylogenies for studies of continuous trait evolution, caution must be taken with ecological traits that are correlated to sampling probabilities, a scenario that caused biased estimates in my simulation. This includes traits tightly linked to rarity, such as abundance and range size (*Gaston, 1994*), as well as niche axes such as habitat type. Rarity is an emergent property of species and not a property of individuals. Traits with potential to bias model-based analyses will likely also be similarly emergent species-level traits. However, it is difficult to generalize about the types of traits that will negatively affect parameter estimates, because they will vary depending on the ecology and life-history of the clade under study. Because ecology and life-history are most well-known by the researchers investigating specific clades, it is advisable that these researchers use their biological intuition to stay mindful of traits that they believe may be linked to rarity and sampling probabilities in their study organisms, especially in situations with extremely high proportions of missing taxa.

The statistical power of a tree and comparative dataset to correctly identify the model that generated the data and to estimate model parameters scales with tree size (*Beaulieu et al., 2012*; *Boettiger, Coop & Ralph, 2012*). For example, trees with up to 128 tips sometimes do not allow the correct selection of an OUM model over OU (*Beaulieu et al., 2012*). The problem of low statistical power in small trees is related to, but distinct from, the problem of missing taxa. Phylogenetic trees may have small sizes and consequent low statistical power for three reasons: (1) because taxon sampling is complete but the clade under study is in fact a small clade; (2) because the clade under study is known to be larger but researchers were unable to sample many known taxa, or (3) because the clade is larger but contains

many undiscovered, undescribed or recently extinct species, so that researchers may not even be aware that their study contains missing taxa. In my simulations, I isolated the missing taxa problem from the tree size problem by keeping the number of tips constant *after* pruning missing taxa, but this is not realistic. In practice the two problems cannot be addressed separately, as under reason 2 above, and may not even be distinguishable, as under reason 3 above. Assessment of statistical power should be standard in comparative analyses. A simulation-based method to accomplish this was described by *Boettiger, Coop & Ralph (2012)*, but it implicitly assumes complete taxon sampling (reason 1 above) and it is unclear if it is adequate in situations with missing taxa. *Slater et al. (2012)* described a method to account for missing taxa in comparative studies, but it assumes that missing taxa are phylogenetically random, a scenario that I argue is unlikely. As far as I am aware, there are no available methods to account for clumped, correlated or unknown (reason 3 above) missing or recently extinct taxa.

In fact, the scenarios produced by my simulations, where I pruned terminal branches from a tree to represent missing taxa, are indistinguishable from scenarios that would be produced by recent extinctions (i.e., extinction at the tips of the phylogeny only). Therefore, my findings might also be interpreted in that context as meaning that random, clumped or correlated recent extinction appear to have little effect on model performance. My results should not, however, be extended to scenarios with extinction in internal branches, which are likely to be prevalent in real datasets (*Stadler, 2010*; *Slater, Harmon & Alfaro, 2012*; *Stadler et al., 2018*). As a case in point, *Slater, Harmon & Alfaro (2012)* found the performance of model selection to significantly deteriorate when randomly-simulated fossil (extinct) taxa were pruned from trees. This deterioration is likely to be even greater if extinction is clumped (*Vamosi & Wilson, 2008*; *Rabosky, 2009*) or correlated (*FitzJohn, 2010*; *Harvey & Rabosky, 2018*) to trait values.

The roster of phylogenetic models of trait evolution is large and ever-growing, and here I examined the effects of missing taxa on only a very small part of that universe. *Rabosky (2015)* found that results for discrete traits were generally similar to those for continuous traits, but before further analyses it is difficult to speculate on how my findings might apply to other models. Furthermore, the scenarios I used to simulate missing taxa were quite limited. Whereas it is clear that missing taxa will often be phylogenetically clumped and/or correlated to the investigated traits, I cannot be sure that the ways in which I simulated those scenarios are optimal. In particular, my non-phylogenetic simulation of corMT is perhaps less than ideal. A more realistic way to simulate that scenario would be under a multivariate phylogenetic model where the value of a liability $L$ is governed both by a Brownian Motion process and by a correlation with a trait of interest $T$, which may be evolving the same or a different model than $L$. However, to the best of my knowledge, that type of multivariate model is not currently available.

Another caveat is that, my simulations notwithstanding, scenarios of missing taxa indisputably exist that do severely mislead phylogenetic comparative methods. For example, without delving into details, a situation in which all taxa at one extreme of the trait value distribution are deterministically missing from a dataset (i.e., all tips with the X% greatest values of $T$ are always unsampled) would truncate the distribution of trait values, altering its

 

mean and variance and likely leading to selection of incorrect models and poor performance of parameter estimates. However, that scenario is unrealistic, because even the rarest taxa will always have a non-zero chance of being discovered, collected and sampled.

As a final cautionary note, my study, as all simulation studies, is only valuable to the extent that our models are accurate. In reality, evolution is likely to proceed in a much more heterogenous manner than reflected in any of our current, simplistic models of trait evolution. We do not know, and might never know, the real generating "models" of empirical datasets. Moreover, actual data, unlike the data I simulated here, are not ideal (e.g., they have measurement error). Therefore, I recommend that my results be not uncritically taken to be applicable whenever any of the models I tested here is fit to real data. The processes that generated a given real dataset may or may not be as robust to missing taxa as are our simple models. Rather, I expect my results to prompt empirical phylogenetic comparative biologists to think critically about missing taxa in their datasets and how they might be distributed. The better the fit of a dataset to a model is, the more likely it is that my findings on the effects of missing taxa will be applicable.

## CONCLUSION

My results demonstrate that realistic scenarios of missing taxa do not affect the performance of selected simple models of univariate continuous trait evolution, except for parameter estimates in situations with a very high proportion of correlated missing taxa. Understanding the impact of missing taxa in comparative methods is a multifaceted problem that still demands much additional work. Topics that are ripe for further study, considering my results, include the effect of missing taxa on models of discrete trait evolution, exploration of better ways to simulate correlated missing taxa, examinations of the level of phylogenetic signal in missing taxa in real datasets, and disentangling the problem of small tree size from the problem of missing taxa. The framework I have provided here will be a starting point for such studies. To facilitate that, R scripts to replicate my simulations are available on GitHub. Proponents of existing and novel models of trait evolution will benefit by using this framework to more explicitly and thoroughly investigate model performance under realistic scenarios of missing taxa.

## ACKNOWLEDGEMENTS

This study benefitted from feedback from Jeremy Brown, Robb Brumfield, Jake Esselstyn and all participants in the Phyleaux discussion group at LSU. All simulations were run in the Odyssey computing cluster at Harvard University, access to which is thanks to Scott Edwards.

### Funding

This research was supported in part by NSF grant DEB-1146265, and by a "Science Without Borders" doctoral fellowship to Rafael S. Marcondes from Brazil's National Council for

Scientific and Technological Development (CNPq; 201234/2014-9). The funders had no role in study design, data collection and analysis, decision to publish, or preparation of the manuscript.

### Grant Disclosures
The following grant information was disclosed by the author:
NSF: DEB-1146265.
Science Without Borders.
Brazil's National Council for Scientific and Technological Development: CNPq; 201234/2014-9.

### Competing Interests
The authors declare there are no competing interests.

### Author Contributions
- Rafael S. Marcondes conceived and designed the experiments, performed the experiments, analyzed the data, contributed reagents/materials/analysis tools, prepared figures and/or tables, authored or reviewed drafts of the paper, approved the final draft.

### Data Availability
R scripts to replicate the analyses and results of the simulations are available at https://github.com/rafmarcondes/Missing_taxa.

### Supplemental Information
Supplemental information for this article can be found online at http://dx.doi.org/10.7717/peerj.7917#supplemental-information.

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
