# Peer review of "Realistic scenarios of missing taxa in phylogenetic comparative methods and their effects on model selection and parameter estimation"

_PeerJ, doi:10.7717/peerj.7917_

## Round 0.1 · original submission · Major Revisions

Dear Dr. Marcondes and colleagues:

Thanks for submitting your manuscript to PeerJ. I have now received three independent reviews of your work, and as you will see, one reviewer recommended rejection (R1), while the other two suggested substantial revisions. I am affording you the option of revising your manuscript according to all three reviews, but understand that your resubmission will likely be sent to at least one new reviewer for a fresh assessment.

The reviewers raised many concerns about the manuscript. Please note that reviewer 1 has provided a marked-up version of your submission. All of these concerns need to be addressed, especially those regarding the experimental design, simulations and problems with references. There seems to be a strong agreement across the reviewers regarding these problems.

Therefore, I am recommending that you revise your manuscript accordingly, taking into account all of the issues raised by the reviewers. I do believe that your manuscript will be closer to publication form once these issues are addressed.

Good luck with your revision,

-joe

Reviewer 1 ·

Basic reporting

there are many mistakes in references.

Experimental design

no comment

Validity of the findings

no comment

Additional comments

there are many mistakes in references.

Annotated reviews are not available for download in order to protect the identity of reviewers who chose to remain anonymous.

Reviewer 2 ·

Basic reporting

# "Clear and unambiguous, professional English used throughout."
Yes. The article is very well written.

# "Literature references, sufficient field background/context provided."
Partly.
- The article is well referenced, and sufficient context is given in the introduction on comparative methods.
- This is a simulation study, but no details are provided regarding the specific simulation tools and methods used. For instance, from the provided R script, I can see that packages "ape" and "castor" are used, respectively, for tree generation and MK model evolution, but they are not cited.
- No comparison is made with previous simulation studies on similar topics. For instance, Cooper et al. 2016 could provide a comparison point for the base case with no missing values.
- Nothing is said about multivariate comparative methods. It might be good to specify at the beginning that this study is limited to univariate methods, as this confused me a little bit at first.
- No reference is made to the classical work of Rubin 1976 (see also Little and Rubin 2002) on inference with missing data. This could be useful, as the "phylogenetically clumped" scenario could actually be MAR (Missing At Random, it does not depends on the trait value), which might explain why it has little impact on the inference. On the other hand, the "correlated" scenario seems NMAR (Not MAR, depends on the trait value), and is typically a case where ignoring the source of missing values might impair the inference, as shown in the analysis.


# "Professional article structure, figures, tables. Raw data shared."
Partly.
- The article is well structured, and easy to follow.
- The scripts to produce the simulation are provided (but not the generated data actually used for the figures).
- Table 1 provides a nice summary for the various conditions.
- This is maybe a matter of personal taste, but could table 2 be transformed into a plot, at least in the main text ? Then the trends might appear more clearly.
- Same remark for table 3.
- Figure 1 is informative. A few comments:
* Please precise in the legend the percentage of missing values.
* Would putting an actual example of the simulation (with 300 species) really impair the illustrative power of the figure ?
* Also, Fig.2C-D might be misleading, as it seems that the "blue" regime does not really represent "no fewer that 25% [...] after dropping the tips representing missing taxa." (l.190-191). From this figure, I have the impression that almost no tips from the "blue" regime are left, which might explain the high bias on theta_1.
- I find Figure 2 not really informative:
* Maybe add facets for each of the scenarios, so that the reader knows what to compare. You could even repeat the base scenario with no missing data to give a comparison in each case.
* No information is provided on the legend on the two numbers of each plot.
* Instead of putting numbers, maybe you could draw points, with lines linking them together to show the trend better ?
* In order to compare the different parameters better, it might be good to *normalise* the bias scores. This is an important remark in my opinion, because theta is around 10, while sigma^2 is around 0.5, so the raw bias on the small parameter might appear smaller, but the scaled one is actually similar, if not larger. This could change the interpretation of the figure given in the text.
* This could be a matter of taste, but maybe violin plots would be easier to read than the dotted empirical distribution.
* Maybe rank the rows according to the model, not the parameters (as they do not mean the same thing in different models anyway). Then maybe show a visual separation between each model's block of parameters.

# "Self-contained with relevant results to hypotheses."
Yes. The article is well focussed, it clearly states the hypothesis, conduct the analyses, and then draw conclusions on these.

Experimental design

# "Original primary research within Aims and Scope of the journal."
This is an exploratory simulation study. I don't feel competent to asses whether this is within the "Amis and Scope" of the journal, and leave that to the Editor's judgement.

# "Research question well defined, relevant & meaningful. It is stated how research fills an identified knowledge gap."
Yes. The research question is well identified, and relevant to the field.

# "Rigorous investigation performed to a high technical & ethical standard."
Partly.
* Simulation scenarios:
- The simulation scenarios are well described, easy to follow, and seems rigourous.
- I really appreciate that the selection strength is given as a phylogenetic half-life, as this is very informative (and not always well specified in other studies).
- Tree generation: as shown in the scripts, the trees are generated with a birth-death process, and then rescaled to unit high, which is different from simulating them directly conditionally on their age (see e.g. function `sim.bd.ntaxa.age` in TreeSim, Stadler 2011). This could have an impact on the tree shape, and hence on the result, so it might be important to mention for comparison / reproducibility reasons.
- Regimes generation: please justify the use of a MK model here. Could the fact that the minor regime has a varying percentage of tips in different analyses bias the results ? In particular for the "correlated" scenario, since tips with high values are typically in the minor regime. See previous remark on Fig.2.
- Latent trait L: maybe give the variance of the BM used.
- corMT scenario: maybe give more precisions on the way the probabilities are chosen. How do they vary with the trait exactly ? Does the fact that the mean probability is the desired proportion of missing data ensure that you actually get the right number ?
- Minor note: notation "T" is usually used for a "Tree", so this confused me a little. But that might be a matter of personal taste.
* Scores used:
- AICc score: please give a citation for the score. Also, it might be good to give a comparison point to what is a "low" delta AICc.
- Bias: as mentioned previously, I think it would be more informative to normalise the bias with the true parameter value, unless there is a good reason not to.
- Precision: maybe give the formula, or more details ? I could not see whether this one was normalised or not.
* Results
- 30% error rate in the BM case seems quite high at first sight. If I'm not mistaken, Cooper et al. 2016 find in similar scenarios (with no missing data, and no measurement error) an error rate that is not above 10% (see their table 2).
- AICc scores: why is less than 2 good ? See remark above. Maybe a citation would help.
- bias: see remark above about the normalisation. E.g., for the OUM, the bias on theta_1, as stated in the text, goes from -0.254 to -0.454. For the same scenarios, the bias for sigma_0 goes from 0.025 to 0.036. But the true parameters are, respectively, 11 and 0.5, so the normalised bias are -0.023 and -0.041 (theta_1) and 0.05 and 0.072 (sigma_0). So the relative bias on theta_1 is multiplied by 1.8, which is about 0.02 units, and the bias on sigma_0 is multiplied by 1.4, which is about 0.02 units too. One could argue then that the two parameters do not behave that differently, and that sigma_0 in not much more robust to bias. But maybe I am missing something, or misunderstood a point ?
- For theta_1, could the high bias be explained by the fact that this regime has fewer tips, and that they are more affected by the correlated missing values, since they are higher ? See previous remark about Fig.2.
- Results on precision are not interpreted.

# "Methods described with sufficient detail & information to replicate."
Partly.
- Methods are described in good details, and the procedure is very clear.
- See section above for some minor caveats on the scenarios.
- Functions used for simulation are provided in two R scripts. However, the master script to call these function and produce the output is not given.

Validity of the findings

# "All underlying data have been provided; they are robust, statistically sound, & controlled."
No actual data. But see above: functions are provided, but not actual scripts, nor the data generated and used in the article.

# "Conclusions are well stated, linked to original research question & limited to supporting results."
Yes. The discussion and conclusion are well suited to the analysis, draw appropriate interpretations. Some comments:
- l.279-290: Maybe a reference to Rubin 1976, as previously mentioned, could help understand this point.
- l.291-297: See previous remarks on normalised bias for sigma^2 and alpha, that might change the interpretation here.
- l.298-314: I'm not sure I understand fully the point about BDPR. Is the message that it's better to use a tree with missing data, than one with inferred data ?
- l.259-363: I don't understand the "multivariate" latent trait approach. Maybe I missed something, but could you explain a bit further what you would do, and why it would be better ?
- l.364-372: The scenario you describe, although not realistic, would be really easy to simulate. Maybe it could be worth to include it in the analysis ? Also, the interpretation of the correlated scenario is highly dependent on the way the probabilities are generated (see remark above). Maybe you could include several functions to do that, giving a more or less high "missing probability gap" between the low and high values ?

Additional comments

Overall this is a well written paper, that is well focussed on a specific research question, and try to address it through simulations. If well conducted, those analyses could be of good use in the comparative biology field. However, in addition to the technical points raised above (that could change the interpretation of the results), I think this paper mostly lacks references, both on the statistical side (MAR/NMAR sampling scheme) and on the simulation side (comparison with previous simulation schemes).

Reviewer 3 ·

Basic reporting

Excellent writing. Good reference to literature but there are a couple additional citations that should be included (especially Ho and Ane 2014, see below).

Experimental design

Good to go.

Validity of the findings

I have no problem with the validity of the findings.

Additional comments

I commend the author on pursuing an area of strong interest to biologists interested in phylogenetic comparative methods (PCM), and in evolution more generally. This manuscript was extremely clearly written, especially in the introduction. I thought the author did a great job of explaining the need for this kind of study in terms that should be digestible by biologists with a variety of areas of expertise. I was very interested in understanding the outcomes of the simulation studies.

As I found the simulations to be valuable, and to highlight some important areas of concern for PCM, my criticisms mostly pertain to interpretation (and presentation) of the results.

1) The simulated data were generated under simple evolutionary models, in which the true model is known. The author has added a dose of realism with respect to sampling (and has also mimicked recent extinction - see below). Thus the author has found that somewhat standard methods used to distinguish between evolutionary models are in a variety of cases not misled under more realistic sampling regimes, but that parameter estimation suffers exaggerated bias in some cases under these more realistic sampling regimes. As the data are still ideal data despite incomplete sampling, I interpret the main results as highlighted (that statistical inference regarding evolutionary processes tends not to be misled in model selection across these cases) as a demonstration that the statistical inference framework can clear a pretty low bar. I agree with the author that these results demonstrate that particular statistical inferences are robust to incomplete sampling for ideal data, where the model is known. However, as evolution does not in reality proceed according to these simple models (e.g. evolutionary processes are likely far more heterogeneous), as we don’t know the generating model, and as empirical data are not ideal (e.g. we have measurement errors), as a biologist interested in using PCM for empirical data I can only take so much confidence from these results. As the author points out in the discussion, we don’t always know the trait values of taxa that are missing from such analyses (especially true of extinct taxa), and thus we can’t know if, for example, we’re missing taxa that belong to a different OU regime.

Consequently, I think the most valuable results from the simulations are those showing where we could be misled in the inference process, even for ideal data. I wanted to know more about how we are misled. E.g., what explains the systematic downward bias in the estimation of theta1 in the OUM models? Is it related to the “intrinsic inference difficulties” in OU models raised by Ho and Ané 2014 Methods Eco Evo? Are the biases likely to be worse for higher theta values? Also, what explains the inference issues when BM is the model generating the data? Has this issue been noted before, e.g. in Beaulieu et al 2012? Which models receive better AICc scores when the generating model does not have the best AICc score? I wondered if you might include an additional table or figure showing which models were preferred by AICc when the generating model was something different.
The discussion touched only briefly on the results, and pivoted to discussions of other issues in the literature, including BDPR. I felt it would be more relevant to put your simulations into the context of previous work on inference difficulties for continuous trait evolution – the discussion regarding sample size was interesting.

2) The manuscript and simulation study addresses studies of extant taxa in large
clades. Extinction looms large here. Sampling for all studies of extant taxa in large clades is incomplete because of it. But the consequences of extinction are first considered at the very end of the discussion. I’d suggest that you devote some space in the intro, discussion, or both to the topic of extinction, in part because you’re mimicking recent extinction by removing tips. There are extinction scenarios that are likely similar to your “clumped” and “correlated” missing taxa scenarios. For example, studies of extant birds only would severely underestimate the number of transitions to flightlessness, as compared to studies including recently extinct taxa (this is discrete obviously, but the bias would likely impact studies of continuous traits like body mass or size-corrected wing length). You might cite papers using e.g. the fossilized birth-death model in studying traits, and peruse some more of the literature for relevant papers.


Some minor issues:

Abstract:
I think you’ve understated your case here when you say that “often the tree and trait dataset contain missing taxa.” I suggest re-phrasing to say that these analyses have rarely, if ever, contained all taxa within a clade. This is most obvious when considering extinction, which occurs even in young clades and results in incomplete sampling.


Lines 111 – 112: I’d add in a phrase to indicate you’re saying these things are heritable at the lineage level, just to help people who don’t usually think about heritability at the macroevolutionary scale.
Line 167: Suggest rewording from “It is worthy to reiterate” with “I reiterate”
Line 232: sample-size corrected?

Lines 287 – 290: I think this interpretation relies on large trees and relatively homogeneous evolutionary processes (which are true for your simulations but won’t be for a great deal of empirical data), and that merits mentioning here. Having phylogenetically clumped missing taxa in a smaller tree or where processes are more heterogeneous could have consequences for model selection problems and parameter estimation.

Lines 307 – 309: Jetz et al 2012 published both genes-only and BDPR trees, which made me wonder whether Rabosky 2015 needed to prune the BDPR trees for that study?

Lines 320 – 322: I had trouble understanding the end of this sentence. Consider rephrasing for clarity.

Lines 362 – 363: You might examine whether it would be possible to do what you’re describing using the “phenotypic tango” model, Arnold and Houck 2016 Am Nat.

Lines 370 – 372: To me, this caveat about extinction is really important, to the point where the scenario is really not unrealistic at all.

Table 1: I’d re-phrase the description of L at the bottom of the table for clarity.

The end of the Table 3 legend appears to be truncated. (oh – it’s on the next page so there’s a floating Table 3 legend). The figure 2 legend in the manuscript I read was truncated.

---

## Round 0.2 · Major Revisions

Dear Dr. Marcondes and colleagues:

Thanks for resubmitting your manuscript. I have only been able to receive one review from the previous three reviewers. This reviewer is very pleased with your revision, but still has some concerns. I find them valid, thus I am offering you the chance to revise your work a second time. I do believe your work will be close to publication after addressing these issues.

Good luck with your revision,

-joe

Reviewer 2 ·

Basic reporting

See below.

Experimental design

See below.

Validity of the findings

See below.

Additional comments

# General Remarks:

I thank the author for his answers to my questions, and the editor for this opportunity to review the revised version of the manuscript. I feel that the manuscript is now stronger, and easier to read. However, there are still a number of points that, in my opinion, remain problematic, as summarised below (see more details in the line by line comments).

- Figure 3 is a major limitation of this article in my opinion. Not only is it very hard to read, but it might also be misleading. Since it is one of the main output of this simulation study, and central to the paper, I would strongly advise for a major revision of this figure before publication. This is a key point, because it drives all the interpretations that are made of the results (see various points below).

- Discrepancies between these results and previously published ones are stated but not explained. I find this lack of explanation troubling, as it could indicate, at best, a design difference that is worth noting, or, at worst, a flaw in one of the studies. In my opinion, finding the roots of this discrepancy is essential in this work.

- I found that the statistical notions about missing data were presented in a confusing way, and as such did not help the interpretation of the results. I would suggest either clarifying this, or completely remove it.

# Reproducibility and additional script on GitHub

On the version I consulted (dated June 12th on GitHub, and most recent when last consulted on July 10th), it is said:
"I freely admit that my scripts are poorly-annotated, clunky, innefficient and just plain ugly."
I understand this remark might have been made with a humorous goal, but is that really the standard we want to ensure reproducibility ? Reading this caveat, I refrained from inflicting a careful review of this material on myself.

Just browsing through the file, I found a few inconsistencies, e.g. one of the main functions ("missing_data_corr_function") is never called in what appear to be the master script (named "run_simulations.R"), while some un-provided files are sourced (e.g "remove_rows_with_all_nas.R" in "sumarize_results").

(Permanent link to the version used: https://github.com/rafmarcondes/Missing_taxa/tree/348c7b738974f767ab8609e50baf7374d4ecd1c6)

# Detailed comments

(Line numbers refer to the "peerj-37170-tracked_changes.docx" file.)

## l.94-104 and 386-388
- I think there might be a small confusion here on the concept of types of missing data. Data is missing completely at random (MCAR) if missingness does not depend on the value of the data, and is missing at random (MAR) if it only depends on observed components, and not on the components that are missing.
- For a rigorous introduction, see e.g. Section 1.3 (p.11) in Little, Rubin, Statistical Analysis with Missing Data, Second edition, 2002, John Wiley & Sons, Inc., Hoboken, New Jersey.
- Usually, if the data is MCAR or MAR, ignoring the sampling mechanism will yield unbiased estimates. However, if it is NMAR (non MAR), then ignoring this mechanism will induce some bias.
- I mentioned it because it seemed to fit nicely to interpret some of the results presented here. However, I did not write down everything, so this is just an intuition, that might be irrelevant. Using this terminology implies precise meaning, that needs to be carefully checked (in particular with respect to the status of the trimmed of complete tree as data). I would suggest that the author mention this literature only if he feels confident that this is indeed relevant.
- In the cited study (Ingram & Mahler 2013), isn't the data even MCAR ?

## Table 2
If space is an issue, I would suggest to remove or move to the supplementary, as the information is redondant with Fig. 2.

## Figure 1
This is just a suggestion: maybe delete (D), but colour the branch of each of the trees in (A-C) according to the regimes ? That way the impact of the regime switch on the trait value and missing data might be seen more clearly.

## Figure 2
In each bar plot, it might be helpful to highlight the colour corresponding to the true model (e.g. by shading or fading the other colours). That way the proportion of "right answers" would appear more clearly.

## Figure 3
* I agree that this figure is key to the paper. However, I disagree that the goal of the figure is to present "the rawest possible depiction of the results". The interested reader can always go look at the raw results (in the supplementaries), but the figure should help him or her understand the results. As such the figure is really hard to read, and could even be misleading (see below). Choices are made anyway in the way to plot and order the data, and I think that there is much room for improvement.
* I still think that adding facets for each scenario would help, precisely to enable a "top-to-bottom looking".
* The trend does not appear clearly, maybe a line would be helpful.
* Raw numbers in the figure are very hard to interpret. I still think a visual representation of them would be better. (Tools like box plots or violin plots precisely being typical ways of showing the dispersion of the estimates graphically.)
* Bias numbers are now normalised, but not the representation of the results. This discrepancy could be misleading. Taking the same example as previously, the normalised bias on theta_1 and sigma_0 increase by the same amount (0.02, for OUM corMT), but on figure J the drop looks much bigger that on figure D, because the raw values are represented.

## Regime size
Thank you for the extra figures and explanations.
However this was not exactly my question, sorry if I was unclear.
In the paper, the case is being made that the bias on theta_1 for OUM corMT is the one with the most significant change. Although this can be nuanced when using the normalised bias (see other remark), my question was, could this drop be also partly imputed to the mere drop of tip observed in this particular regime ? If I understood correctly, under the corMT scenario, missing values will tend to be concentrated on high values, which will tend to depopulate regime 1, and hence increase the bias on theta 1.
Would that interpretation make sense ?
If it does, I don't think it shows a flaw in the simulation design. However it could potentially be used to nuance the claims about the drop in performance for OUM in the corMT scenario.

## l.263-282
- I'm not sure I understand what is meant by "relative probability". Those numbers do not seem to sum to one.
- I'm still unclear about the sentence: "R native function sample.int, which rescaled it to have a mean equal to my desired proportion of missing taxa and used the rescaled vector to sample tips to be dropped from the tree". Which mean does this refer to ? For the help of the R sample function, I have: "The optional prob argument can be used to give a vector of weights for obtaining the elements of the vector being sampled. They need not sum to one, but they should be non-negative and not all zero. [...] If replace is false, these probabilities are applied sequentially, that is the probability of choosing the next item is proportional to the weights amongst the remaining items. The number of nonzero weights must be at least size in this case." From what I understand, the right proportion of missing data is obtained because the right number of sampled integers is passed to 'sample.int' (specifically, 'PMISSING*nrow(tipdata)' l.108 of file "correlated_MT.R"), not because of a rescaling of the mean. But maybe I misunderstood something ?
- In the rebuttal letter it is explained that "However, that is not true for the rMT (random MT) scenario, where the actual proportion of missing taxa is probabilistic." Maybe I missed it, but I could not find a mention of that in the manuscript. Why is the proportion probabilist in this case ? It might be worth explaining this in the description of the simulations.

## l.290
Please provide a citation for the definition of AICc.

## l.336
Burham & Anderson (2004) cannot be found in the references at the end.
Also, is it delta AIC, or delta AICc ?

## l.346
0.2 -> 0.02 ?
A lot of time is spent discussing small differences, but maybe it should be stressed that there is a factor 10 between the biais on alpha and all the other parameters.

## l.364-367
How is the drop for theta_1 from -0.0231 to -0.0413 worse than the one from -0.007 to -0.055 for sigma_1 BMS, or from -0.0032 to -0.0374 for sigma_0 BM ? I might be wrong, but I think Figure 3 is misleading on this, or not carrying the proper information (see remarks above).

## l.413
"I have no compelling interpretation to explain this discrepancy between mine and Cooper et al.’s (2015) studies."
I find this statement a bit worrisome. If there is no design difference (in the simulation, or in the inference) that one could point out to explain the difference, then couldn't one worry that one of the results is misleading ?
If this comparison is included in the paper, I think it is essential to provide at least an intuitive explanation of why they differ.

## l.521
"then L" -> "than L" ?

---

## Round 0.3 · Minor Revisions

Dear Dr. Marcondes and colleagues:

Thanks for resubmitting your manuscript. The one reviewer from the previous round has again provided a review. This reviewer is very pleased with your revision, but still has some concerns. I find them valid, thus I am offering you the chance to revise your work a third time. I do believe your work will be close to publication after addressing these issues.

Good luck with your revision,

-joe

Reviewer 2 ·

Basic reporting

See below.

Experimental design

See below.

Validity of the findings

See below.

Additional comments

# General Remarks

I thank the editor for this opportunity to review this manuscript, and the author for answering my questions. I think that the most problematic issues previously pointed out have been addressed, and that this manuscript could be suitable for publication.

However, and as a personal opinion, I should say that, despite the absence of compelling obvious flaws, I have a rather low confidence in the work presented here, and I think it could improve on the three main following points: 1- Proper formatting of the code; 2- Informative representation of the results; 3- Integration with the existing literature.

I explain these points in more details below. However, I would perfectly understand that they could be taken as opinions rather than actual limiting flaws. As illustrated by the author's recent witty tweets on my previous review (yes, twitter is a public place, and grumpy reviewers can also read what's on it), we might not share the same vision on what an informative graph should look like. I would hence recommend the editor to take an extra opinion from another researcher, and I leave the decision to his discretion. Whatever his decision should be, I will not review any further version of this manuscript.

1- Proper formatting of the code.

Thank you for trying to clean the code between the previous version and this submission. I appreciate your effort, and I think clear and reproducible code is essential in the modern publication process, especially for such a study relying essentially on simulations.

Browsing the code, I still found it somewhat hard to read. It might benefit from a more rigorous formatting, based on a commonly accepted style (see e.g. https://style.tidyverse.org/). Applying the "lintr" package (see reference on the webpage above), I for instance found 767 notes and warnings on the file "correlated_MT", which is only 320 lines long, making for more than two per line. I'm not saying this shows a fatal flaw in the code, but this does illustrate the fact that this code might be hard to read for a human, which makes it more error prone.

This could extend to the use of base R functions themselves, as illustrated by the previously somewhat confusing explanation of the sampling procedure (see also comment below).

2- Informative representation of the results.

I discussed this point at length in my two previous reviews. I still think that this paper lacks one figure presenting the results in a clear way, summarising the main important features so that the reader could get a quick and clear grasp of the main trends. I just think that tables and raw data points are hard to interpret, and some inaccuracies in the exposition of the results in the main text by the author in previous versions tend to demonstrate that I might not be completely wrong.

This would not prevent the author to give the full results in whichever form he likes. I would however suggest to put the easy-to-read summary in the main text, and the more detailed, raw-data information in the appendix. But, again, that might just be a matter of personal preference.

3- Integration with the existing literature.

I have also already discussed this point, and the apparent inconsistency with some of the results existing in the literature. The general reason for this discrepancy given by the author (AIC vs LRT) could be a valid point, but it is quite broad, and not backed up either by literature or data. It made my strong suspicion go away, but I'm still not entirely convinced (see detailed comments below).

# Detailed Comments

(Line numbers refer to the "peerj-37170-trackedchanges_31july2019.docx" file.)

## l.262
Why do you apply a different sampling method for rMT and corMT, leading to a non-fixed number of missing tips in the rMT case ? Instead of using "sample" on a binary outcome with a probability equal to the percentage of missing (l.132 of correlated_MT.R), it seems that one could just apply again "sample.int" with the correct size (as in l.127), but with all probability weights equal to one (or prob=NULL). But maybe I am misunderstanding something ?
I agree that averaged over 1000 simulations, this should not have a strong impact, but I'm wondering whether there is a reason for the use of a different method in the two cases.

## l.422
I agree that the discrepancy could be due to the use of a different score (AICc instead of likelihood ratio test). This would be easy to check. I would suggest including the likelihood ratio test for model selection as an extra analysis for the sake of comparison, at least in the supplementary. That would be reassuring, and would easily settle this discussion. Also, for the comparison between AIC and LRT, the following reference might be of interest:
Lewis, F., Butler, A., & Gilbert, L. (2011). A unified approach to model selection using the likelihood ratio test. Methods in Ecology and Evolution, 2(2), 155–162.

## Figure 3
It's fine if the author does not like violin plots, but I still think that at least adding a summary statistic (e.g. a boxplot) to the raw data could be useful, so that the reader could follow easily the main trends. I also still think that separating different scenarios by panel could help. But see my above general comment. Plotting the normalised estimates, instead of the raw ones, could also help grasping the trends better.

## Table 4 and 5
Replacing, or at least complementing, these tables by graphs could help the reader to see the main point. But, again, that might be a matter of personal preference.

# Typo

## l.256
Missing period after the formula, or extra capital letter.

---

## Round 0.4 · accepted · Accept

Dear Dr. Marcondes:

Thanks for re-submitting your revised manuscript to PeerJ, and for addressing the concerns raised by the reviewer. I now believe that your manuscript is suitable for publication. Congratulations! I look forward to seeing this work in print, and I anticipate it being an important resource for the field of phylogenetics.

Thanks again for choosing PeerJ to publish such important work.

-joe